

# Development and validation of COVID-19 vaccination perception (CoVaP) instrument among healthcare workers in Malaysia

Siti Nur Aisyah Zaid[1,2], Azidah Abdul Kadir[1,2], Mohd Noor Norhayati[1,2], Basaruddin Ahmad[2,3], Muhamad Saiful Bahri Yusoff[1], Anis Safura Ramli[4] and Jasy Suet Yan Liew[5]

[1] School of Medical Sciences, Universiti Sains Malaysia, Kubang Kerian, Kelantan, Malaysia
[2] Hospital Pakar Universiti Sains Malaysia, Kubang Kerian, Kelantan, Malaysia
[3] School of Dental Sciences, Universiti Sains Malaysia, Kota Bharu, Kubang Kerian, Malaysia
[4] Faculty of Medicine, Universiti Teknologi MARA, Sungai Buloh, Selangor, Malaysia
[5] School of Computer Sciences, Universiti Sains Malaysia, Gelugor, Penang, Malaysia

Corresponding author
Azidah Abdul Kadir,
azidahkb@usm.my

## ABSTRACT

**Background**. Healthcare workers (HCWs) play an essential role in facilitating coronavirus disease 2019 (COVID-19) vaccination, and their confidence in vaccination is crucial. Nevertheless, valid instruments for assessing the HCWs' perceptions of COVID-19 vaccination were lacking. This study aims to develop and validate the COVID-19 Vaccination Perceptions (CoVaP) instrument among HCWs in Malaysia.

**Methods**. A literature review and discussion with research teams were conducted to identify the content to be considered. The instrument was developed in Malay language and underwent back-to-back translations to the English version. The initial CoVaP instrument was unidimensional with 12 items. The Malay and English versions underwent a content validation process by seven expert panels. However, only the Malay version underwent face and construct validation. Face validity was assessed using 30 HCWs. The construct validation was conducted in a two-step process using data from two cross-sectional study samples, including 125 and 300 HCWs for exploratory factor analysis (EFA) and confirmatory factor analysis (CFA), respectively. It was a self-administered questionnaire, and the data were collected using both face-to-face and online platforms. The data were analysed using Analysis of Moment Structure version 28.0 and Statistical Packages for the Social Sciences version 26.

**Results**. The analysis showed excellent content (item content validity index (I-CVI) = 0.83 to 1.0, average content validity index (S-CVI/Ave) = 0.97) and face (item face validity index (I-FVI) = 0.87 to 1.0, average face validity index (S-FVI/Ave) = 1.02) validity. The EFA analysis revealed seven items with two domains. Subsequent analysis using CFA demonstrated a two-factor model of seven items with an acceptable level of goodness of fit indexes (comparative fit index = 0.999, Tucker-Lewis index = 0.999, incremental fit index = 0.987, chi-squared/degree of freedom = 1.039, and root mean square error of approximation = 0.011). Finally, the Cronbach's alpha was satisfactory for both domains (0.899 and 0.815).

**Conclusion**. The CoVaP instrument is a valid and reliable tool for measuring perceptions of COVID-19 vaccination among HCWs. The high validity and reliability of the

CoVaP instrument underscore its utility in capturing the unique cultural and contextual factors influencing vaccine perceptions among Malaysian HCWs. Understanding these factors is essential for designing effective public health interventions to address vaccine hesitancy and improve vaccination rates.

## INTRODUCTION

The coronavirus disease 2019 (COVID-19) has caused a substantive impact on social, psychological, economic and health aspects, as well as global mortality (*Bandyopadhyay et al., 2020*). The healthcare workers (HCWs) are one of the vulnerable group to be affected psychologically, due to work related stress (*Hamdan et al., 2023*). The catastrophic consequences associated with the COVID-19 pandemic have accelerated the process of COVID-19 vaccination at an unimaginable speed to keep the outbreaks under control. The COVID-19 vaccination has substantially altered the course of the pandemic, saving tens of millions of lives globally and safeguarding economies from continued disruption and damage (*Watson et al., 2022*).

Currently, the COVID-19 disease has reached a post-pandemic stage and two public health issues have arisen: there is a risk of a new strain emerging and causing another major catastrophic outbreak and the current burden of COVID-19 infection needs effective preventive measures (*Watson et al., 2022*). COVID-19 vaccination remains an effective measure to control this disease and has significant benefits despite the emergence of new variants (*Kitano et al., 2023*). Nevertheless, the rapid development of vaccines, the emergence of new variants, and the dissemination of misinformation relating to the risk of vaccination and policy changes fueled by digital platforms have promoted COVID-19 vaccine hesitancy globally (*Larson, Gakidou & Murray, 2022*).

Vaccine hesitancy is defined as people who delay or refuse vaccination despite its availability, reject or postpone certain vaccines, refuse all vaccines, and accept but remain concerned with it (*Edwards & Hackell, 2016*; *MacDonald, 2015*). The Strategic Advisory Group of Experts (SAGE) on Immunization describe vaccine hesitancy as a behaviour shaped by factors including confidence, complacency, and convenience. Confidence refers to trust in the vaccine and the provider, complacency involves the perceived need or value of the vaccine, and convenience covers accessibility and logistical concerns (*Larson et al., 2014*).

COVID-19 vaccine hesitancy is influenced by a complex interplay of psychological, social, and contextual factors, which have been systematically analyzed using theoretical frameworks such as the 5C model (*Rancher et al., 2023*), the 7C model (*Araujo-Chaveron et al., 2024*), the vaccine hesitancy scale (VHS) (*Akel et al., 2021*), and the vaccine conspiracy beliefs scale (VCBS) (*Al-Sanafi & Sallam, 2021*). The 5C model identifies five key factors: confidence, complacency, constraints, calculation, and collective responsibility (*Rancher et*

*al., 2023*). The 7C model expands on this by adding compliance and conspiracy. *Geiger et al. (2022)* and the vaccine hesitancy scale (VHS) measures hesitancy through dimensions like lack of confidence and perceived risks (*Shapiro et al., 2018*). The vaccine conspiracy beliefs scale (VCBS) assesses the impact of conspiracy beliefs on vaccine acceptance (*Al-Sanafi & Sallam, 2021*).

These models provide a structured approach to understanding the determinants of hesitancy, including trust in vaccines, perceived risks, misinformation, and sociodemographic influences. These frameworks were widely applied to assess vaccination attitudes across diverse populations, including HCWs. For example, studies using the 5C model revealed that socio-demographic characteristics and drivers of vaccine behaviour, such as high levels of trust and collective responsibility, were key drivers of acceptance among the general public in South Carolina (*Rancher et al., 2023*). The 7C model showed that compliance and conspiracy beliefs were critical in understanding vaccine hesitancy among healthcare workers, parents, and adolescents in France (*Oudin Doglioni et al., 2023*). The VHS highlighted that younger age and women were among the factors associated with low vaccine confidence among the general public in Israel (*Grossman-Giron et al., 2023*). The VCBS model demonstrated that higher levels of conspiracy beliefs were associated with lower vaccine uptake in Poland (*Kowalska-Duplaga & Duplaga, 2023*). These insights helped tailor public health interventions to address specific concerns and improve vaccination rates across different demographics.

The impact of vaccine hesitancy is expected to be greater and pertinent to public health, particularly when involving vulnerable frontline groups such as healthcare workers (HCWs). HCWs played a crucial role in the COVID-19 vaccination campaign by improving health literacy, recommending vaccination, and mitigating the spread of the virus. They were pivotal in educating the public about vaccine safety and efficacy, enhancing health literacy and addressing vaccine hesitancy. HCWs' recommendations significantly increased vaccination rates, as patients were more likely to get vaccinated when advised by their healthcare providers. However, they also posed a risk of transmitting the virus to vulnerable groups, highlighting the importance of their vaccination. Additionally, increased absenteeism among healthcare workers due to illness or quarantine impacts healthcare services. Studies have shown that healthcare workers (HCWs) were a critical group for vaccination studies due to their high exposure risk and their role in influencing public health behaviours (*Blake et al., 2024*; *Hoffman et al., 2022*). Therefore, it is crucial to improve the HCWs' confidence in the vaccine before engaging them in promoting vaccination (*Saddik et al., 2022*).

An extensive umbrella review of studies conducted among HCWs and healthcare students in 2023 found that vaccine hesitancy was mainly influenced by sociodemographic factors (like gender, age, and ethnicity), occupational factors (such as COVID-19 exposure and perceived risk), health factors (including vaccination history), vaccine-related concerns (about safety, efficacy, and side effects), social factors (like social pressure and collective responsibility), distrust (in key social actors and pandemic management), and information factors (such as inadequate information and exposure to misinformation (*McCready et al., 2023*). There were numerous studies regarding the hesitancy, perceptions, and attitudes

of HCWs toward COVID-19 vaccine (*McCready et al., 2023*). However, the available tools specifically validated for HCWs are relatively lacking. Most scales were validated for the general public (*Campo-Arias, Caamaño Rocha & Pedrozo-Pupo, 2023*; *Guad et al., 2021*; *Mejia et al., 2021*; *Reznik et al., 2021*). A few validated scales were conducted for the HCWs. These include coronaphobia scale in Mexico (*Mora-Magaña et al., 2022*), corona anxiety scale in Turky (*Evren et al., 2022*), and COVID19 stigma scale in Syria (*Al Houri et al., 2022*). Most of the studies were done during the pandemic and outside Malaysia, which has different cultures, and norms. Malaysia is often described as a ''multicultural country'' due to its diverse population, which consists of multiple ethnic, cultural, and religious groups coexisting within the same nation. The main ethnic groups in Malaysia are Malay, Chinese and Indians. The multicultural nature of Malaysia plays a significant role in shaping public attitudes and behaviours, including vaccine hesitancy. For example, some Muslims in Malaysia expressed concerns about the halal status of COVID-19 vaccines (*Ismail et al., 2023*). Trust in the government and healthcare systems varies across ethnic groups. Historical experiences and perceived discrimination can affect how different communities view vaccination campaigns. Understanding and addressing these multicultural factors is crucial for designing effective public health campaigns to overcome vaccine hesitancy and ensure broad vaccine coverage in Malaysia.

Malaysia began its vaccination program in February 2021, prioritising high-risk groups such as healthcare workers, the elderly, and individuals with comorbidities. By mid-2022, the country had achieved high vaccination coverage, with over 80% of the total population fully vaccinated and a significant portion receiving booster doses. Malaysia has not implemented a nationwide mandatory COVID-19 vaccination policy for the general population. However, vaccination has been strongly encouraged through various incentives, such as allowing vaccinated individuals greater freedom of movement and access to public spaces during lockdowns. For certain sectors, such as healthcare and education, vaccination has been effectively mandatory. The COVID-19 vaccines are provided free of charge to all Malaysian citizens and non-citizens residing in the country. Malaysia's COVID-19 Vaccination Programme included administering a range of vaccines such as Pfizer (Comirnaty), AstraZeneca (Vaxzevria), and Sinovac (CoronaVac), depending on availability and suitability for different population groups The government has borne the cost of vaccination as part of its National COVID-19 Immunization Program (PICK), which aims to achieve herd immunity and reduce the burden on the healthcare system (*Hamdan, Fahrni & Lazzarino, 2022*).

Malaysia has reported relatively high COVID-19 vaccine acceptance rates among its general population. Surveys conducted during the initial vaccine rollout indicated that approximately 80–90% of Malaysians were willing to be vaccinated. *Kalok et al. (2023)*, *Syed Alwi et al. (2021)* and a recent survey in 2024 indicated that the COVID-19 acceptance rate was 56.5% (*Mohd Anuar et al., 2024*). However, COVID-19 booster vaccination rates in Malaysia remain below 50% among the general public (*Khoo et al., 2024*). In Malaysia, specific data on vaccine hesitancy among HCWs is limited, making it challenging to fully understand the barriers to vaccine uptake within this critical group. A cross-sectional

study involving 380 HCWs at a tertiary hospital in Malaysia during the early phase of the pandemic reported low vaccine hesitancy (*Mahmud et al., 2023*).

This study aims to develop and validate an instrument for assessing the perceptions towards COVID−19 vaccination (CoVaP) in the Malay language specifically for Malaysian HCWs. Accurate and reliable assessment tailored to this population is needed to capture the unique cultural and contextual factors influencing vaccine perceptions in Malaysia, helps design interventions specifically addressing healthcare workers' concerns, informs policymakers and healthcare administrators for effective program design and enables tracking of vaccine hesitancy trends over time.

## MATERIALS AND METHODS

The development and validation of the instrument were conducted in two distinct phases, ensuring a comprehensive approach to creating a reliable and valid tool. The method for the development and validation of the instrument followed the practices advocated in the previous reports (*Ostromohov et al., 2022*; *Tsang, Royse & Terkawi, 2017*). Data were collected as previously described in *Zaid et al. (2024)*.

### Phase 1: development of the scale

Firstly, a literature review was conducted to identify similar validated instruments for assessing the perception, attitude, acceptance or hesitancy toward COVID-19 vaccination among HCWs or the general population. The search strategy was based on a previous review study (*Galanis et al., 2021*). The search was conducted from 1st January to 31st March 2022. However, the research team concluded that there was no appropriate and suitable validated questionnaire available for HCWs in Malaysia and decided to develop a new instrument.

Then, a multidisciplinary steering group was set up comprising four members: one family medicine specialist, one nursing tutor, one community medicine specialist (statistician) and one dental specialist (statistician) to identify and revise the items, and construct, format and determine the instrument length. The instrument was developed in Malay language and underwent back-to-back translations to the English version. The items were developed and modified based on several studies (*Abu Farha et al., 2021*; *Alam et al., 2022*; *Fares et al., 2021*; *Kotta et al., 2022*; *Mejia et al., 2021*; *Mohamed et al., 2021*). The issues considered to be important and relevant to the perception of HCWs' COVID-19 vaccination were found, and included religious issues, vaccine safety and efficacy, and misinformation such as the vaccine containing electronic chips and pharmaceutical conspiracy theories (*Wong et al., 2023*).

A few items relating to the occupational risk of infection in HCWs and the vaccine adverse effects reported by the Ministry of Health were included, as they were different from the instruments developed for the general public. The initial draft of the self-administered instrument was unidimensional and consisted of 12 items in the Malay language. The response has a 5-point Likert instrument (0–4), from strongly disagree to strongly agree. Six items used reverse scoring. The total score ranges between 0 and 48 and a greater value reflects positive perception towards the COVID-19 vaccines.

## Phase 2: validation process

The instrument was validated sequentially, including content, face, construct validity and reliability. The Malay and English versions underwent a content validation process. However, only the Malay version underwent face and construct validation. This is because Malay is the national language and is easily understood by most HCWs at various levels.

### Content validity

Content validity is evaluated by determining the extent to which a particular instrument properly represents or reflects the concept being evaluated for a particular purpose. Establishing content validity is crucial for ensuring the accuracy of assessment questionnaires and should be considered a top priority when creating an instrument. This study utilized a method known as content validity index (CVI), the most widely used method (*Abdul Kadir, Noor & Mukhtar, 2021*; *Yusoff, 2019*). Based on the literature, the recommendation for the number of experts varies from as low as two persons to as high as nine persons (*Yusoff, 2019*). At least five experts are suggested to review the instrument to have sufficient control over the chance agreement (*Zamanzadeh et al., 2015*). In this study, we used seven experts.

The content validity of the items was independently reviewed by a panel of experts ($N = 7$) comprising family medicine specialists ($N = 3$), nursing lecturers ($N = 2$), and community medicine specialists ($N = 2$). The evaluation method was carried out by asking the panel members to rate the relevance of each item using two content validity assessment forms based on a 4-point Likert scale ranging from not relevant (1) to highly relevant (*Abdul Kadir, Noor & Mukhtar, 2021*; *Yusoff, 2019*). One assessment, the item content validity index, is calculated based on the number of experts rating 4–5 for each item divided by the total number of experts. Items with I-CVI $\leq 0.70$ were eliminated, and items with I-CVI between 0.70 and 0.79 were rephrased. The average content validity index (S-CVI/Ave) is determined by dividing the total of the I-CVIs by the number of items. A score of 0.83 is considered to meet the acceptable threshold (*Yusoff, 2019*). The expert panel also were asked to make recommendations for each item. The expert panel assessment was conducted from 11th May 2022 until 21st July 2022.

### Face validity

Face validity was tested with 30 HCWs from Hospital Universiti Sains Malaysia (HUSM) from 1st to 14th August, 2022. Participants rated the clarity and comprehensibility of each item on a 5-point Likert scale (1 = not clear/comprehensible to 5 = extremely clear/comprehensible). The item-face validity index (I-FVI) and average face validity index (S-FVI/Ave) were utilised based on the reported method (*Yusoff, 2019*). Items with an item-face validity index (I-FVI) >0.79 were considered to have good clarity and comprehensibility. Adjustments were made based on feedback, and the revised instrument was used for construct validation. The S-FVI/Ave is calculated from the sum of the I-FVI divided by the number of HCWs ($N = 30$). Adjustments were made in response to the feedback from respondents, and the revised instrument was used for construct validation.

### Construct validity and reliability

Two independent HCWs samples were recruited using a cross-sectional study design; 125 and 300 HCWs were recruited for the exploratory factor analysis (EFA) and confirmatory factor analysis (CFA), respectively. The respondents were Malaysian citizens in permanent or contract positions in the public and private healthcare services in Malaysia and included health professionals, managers, and support workers such as cleaners, drivers, hospital administrators, and district health managers, who could read and comprehend the Malay language.

The samples were recruited using chain-referral sampling by distributing the survey link to the social contact of researchers *via* multiplatform messaging apps such as WhatsApp and Telegram or face-to-face methods. The recipient of the link was requested to distribute the invitation to all their contacts further. Implied consent was uploaded on the first page of the survey link, and subjects who click 'I agree' and complete the survey will be deemed to have consented to participate. The data collection for EFA took place from 20th August to 20th. September 2022 (one-month duration), while the data collection for CFA was carried out from 20th. October to 20th. November 2022 (one-month duration).

## Statistical analysis

A descriptive analysis was performed to describe the summary statistics and examine the distributions of the items. The EFA and CFA were used to assess the dimensionality of the instrument. For EFA analysis, we utilised the principal axis factoring with Promax rotation. The items were retained for further analysis if they have the communality and have factor loadings value of $\geq 0.3$ (*Tavakol & Wetzel, 2020*). This cut-off score also has been used in another study (*Guad et al., 2021*). The Kaiser–Meyer–Olkin (KMO) test was used to determine the sampling adequacy with KMO > 0.5 and Bartlett's sphericity ($p < 0.05$) considered acceptable.

CFA was utilized to validate the factor structure identified in exploratory factor analysis (EFA) and to evaluate the quality of fit indicators for the CoVaP latent construct. The normality of factor scores and the presence of outliers were further assessed using the critical ratio, which compares skewness and kurtosis to their standard errors, along with the Mahalanobis distance. The Mahalanobis distance was used for outlier detection by measuring and ranking the distance of each data point from the center of the data distribution.

Construct validity evaluates the degree to which a scale precisely measures the intended concept. Fit of the models was assessed using the following criteria: normal chi-square per degree of freedom (Cmin/df) < 3 (*P* value of > 0.05), comparative fit index (CFI), incremental fit index (IFI), and Tucker-Lewis index (TLI) $\geq 0.9$ and root-mean-square error approximation (RMSEA) $\leq 0.08$. Additionally, the modification indices (MI) were utilized to assess the quality of fit. A high MI value suggests a significant amount of redundancy in a pair of variables.

The internal consistency of the instrument was assessed by calculating the Cronbach's coefficient value for the entire instrument and each of its sub-instruments using SPSS. A coefficient of 0.7 indicated that the instrument had good internal consistency. CFA analysis

was used to calculate composite reliability (CR) and average variance extracted (AVE), which were manually computed using established formulas from the published literature. The CR of 0.6 and the AVE of >0.5 were considered acceptable to reflect satisfactory internal consistency.

Analyses were performed using SPSS Statistics software version 26.0 (Statistical Package for the Social Sciences, IBM Corp., Armonk, NY, USA) and SPSS Amos software version 28.0.

### Ethical approval

The study protocol received the approval of the Universiti Sains Malaysia Research and Ethical Committee (USM/JEPeM/21100700) and the Malaysia Ministry of Health (NMRR ID-21-02113-ZGG (IIR)). Written informed consent was obtained for the content validity and face validity studies. Informed consent is implied in the cross-sectional study by clicking 'I agree' on the first page of the Google form's survey link.

## RESULTS

### Content and face validity

The data for the final content and face validity are shown in Table 1. The I-CVI (score = 0.57 to 1) and S-CVI/Ave (0.76) scores of the first content validation process did not meet the cut-off level. After revising the items, the process was repeated, and the result showed improvement in the I-CVI (0.83 to 1) and S-CVI/Ave (0.97). For the I-FVI (0.87 to 1) and S-FVI/Ave 1.02, the scores were above the cut-off, indicating that the questionnaire items are important, straightforward, and easy to understand for the HCWs (*Yusoff, 2019*). Several modifications also were made to a few items in response to the suggestions provided by the respondents.

### EFA

There were no missing responses from the 125 respondents. The mean standard deviation (SD) age was 35.7 (8.7) years. The majority of the respondents were female (80%), and from the Malay ethnic. The initial analysis showed good sampling adequacy with KMO = 0.81 and a substantial correlation between items with significant Bartlett's test of sphericity (Chi-square = 615.296, $p < 0.0001$).

The eigenvalues of the initial analysis suggested three factors and the scree plot indicated that two to three factors should be extracted. The analysis continued with two factors which only explained 51.32% of the total variance. Consequently, in the final EFA analysis, five items (A1, A5, A6, A11 and A12) were removed one by one resulting in the final model with KMO = 0.849 indicating that the two-factor structure with seven items was appropriate. The factor loadings and communality loads were between 0.40 and 0.91 (Table 2). No multicollinearity was identified with the correlation between the two factors = 0.67. These findings show that the final EFA results indicated good construct validity, and the researcher proceeded with the CFA.

**Table 1  Content and face validity for CoVaP.**

| Coding | Items | I-CVI | I-FVI |
|---|---|---|---|
| A1R | Saya percaya vaksin COVID-19 dapat melindungi kita daripada jangkitan influenza (flu). <br> *I believe the COVID-19 vaccine can protect us from influenza infection (flu).* | 0.83 | 0.90 |
| A2 | Saya berpendapat vaksin ini melindungi kita daripada komplikasi teruk jangkitan COVID-19. <br> *I think this vaccine protects us from the severe complications of COVID-19 infection.* | 0.83 | 0.97 |
| A3 | Saya berasa keberkesanan vaksin dapat mengatasi risiko kesan sampingan teruk vaksin. <br> *I feel the effectiveness of the vaccine outweighs the risk of serious side effects.* | 1 | 0.90 |
| A4 | Saya percaya vaksin COVID-19 selamat diambil kerana telah diuji secara klinikal. <br> *I believe the COVID-19 vaccine is safe because it has been clinically tested.* | 1 | 1 |
| A5R | Saya berpendapat vaksin ini boleh menyebabkan kesan sampingan buruk jangka masa panjang. <br> *I think this vaccine can cause long-term side effects.* | 1 | 0.87 |
| A6R | Saya kurang percaya kepada laporan kesan sampingan teruk yang dikeluarkan oleh Kementerian Kesihatan Malaysia. <br> *I have little faith in the severe side effect reports issued by the Malaysian Ministry of Health.* | 1 | 0.93 |
| A7R | Saya berpendapat vaksin mempunyai cip elektronik. <br> *I believe that this vaccine contains electronic chips.* | 1 | 1 |
| A8 | Saya percaya vaksin ini perlu diwajibkan kepada semua orang dewasa. <br> *I believe this vaccine should be mandatory for all adults.* | 1 | 0.90 |
| A9R | Saya meragui status halal vaksin COVID-19. <br> *I have a doubt about the COVID-19 vaccination halal status.* | 1 | 0.90 |
| A10R | Saya berpendapat perubatan alternatif (homeopati, herba dan lain-lain) boleh menggantikan vaksin bagi memberikan perlindungan daripada jangkitan COVID-19. <br> *I think alternative medicine (homeopathy, herbs, etc.) can replace vaccines to provide protection against COVID-19 infection.* | 1 | 0.93 |
| A11 | Saya percaya pekerjaan saya menyebabkan saya berisiko tinggi untuk mendapat jangkitan COVID-19. <br> *I believe my job puts me at high risk of getting a COVID-19 infection.* | 1 | 0.97 |
| A12R | Saya terpaksa mengambil vaksin ini kerana tuntutan pekerjaan. <br> *I had to take this vaccine due to my work demand.* | 1 | 0.97 |

**Notes.**

R, reverse scoring, I-CVI, item content validity index, and I-FVI, item face validity index.

**Table 2  Descriptive statistics and factor matrix for initial CoVaP.**

| Items | Descriptive statistics | | | | Factors | | Communalities |
|---|---|---|---|---|---|---|---|
| | Mean | SD | Skew | Kurtosis | 1 | 2 | |
| A1* | 2.70 | 1.16 | 0.30 | −0.77 | −0.18 | 0.32 | 0.13 |
| A2 | 4.19 | 0.70 | −0.27 | −0.93 | 0.84 | −0.35 | 0.82 |
| A3 | 4.02 | 0.81 | −0.44 | −0.42 | 0.84 | −0.21 | 0.75 |
| A4 | 3.94 | 0.89 | −0.80 | 0.72 | 0.79 | −0.22 | 0.67 |
| A5* | 3.08 | 1.01 | −0.39 | −0.29 | 0.20 | 0.23 | 0.09 |
| A6* | 3.10 | 0.83 | 0.04 | 0.62 | 0.28 | 0.45 | 0.28 |
| A7 | 4.21 | 0.79 | −0.67 | −0.24 | 0.64 | 0.25 | 0.47 |
| A8 | 3.76 | 0.88 | −0.56 | 0.42 | 0.55 | −0.24 | 0.37 |
| A9 | 3.69 | 1.02 | −0.41 | −0.37 | 0.74 | 0.40 | 0.71 |
| A10 | 3.79 | 0.96 | −0.17 | −1.01 | 0.69 | 0.30 | 0.56 |
| A11* | 4.10 | 0.84 | −0.87 | 0.53 | 0.36 | −0.16 | 0.16 |
| A12* | 2.95 | 1.22 | 0.01 | −0.94 | 0.37 | 0.14 | 0.15 |

**Notes.**

Extraction Method: principal axis factoring

*Item proposed to be removed.

## CFA

The mean (SD) age of the respondents was 37.3 (8.4) years and 76.7% was female. About 34% were nursing staff and 38% were in administrative positions. The CFA analysis showed that the model achieved acceptable fitness index values, indicating good model fit. The RMSEA of the model (0.011) indicated adequate fit and the goodness of fit index (GFI), CFI, TLI and the normed fit index (NFI) showed a good fit (0.987, 0.999, 0.999 and 0.987, respectively). The Chisq/df was less than 3, indicating adequate fit and acceptable value. Hence, the final CoVaP contained seven items with two factors, as illustrated in Fig. 1.

## Reliability

Table 3 presents the results of factor loadings, AVE and CR for the CoVaP. The reliability of the instrument was assessed based on the values of Cronbach's alpha, AVE and CR. The factor loading value of each item was ranging from 0.57 to 0.85. The AVE result value was more than 0.50, and the CR value was more than 0.60. Thus, the final instrument version showed valid and good convergent validity based on the results of the factors loadings, CR, and AVE, all of which were above acceptable values.

## DISCUSSION

HCWs have a significant role in promoting vaccination and can influence patient or public vaccination uptake. Therefore, it is crucial to improve their confidence in vaccination and engage them in activities targeting vaccine hesitancy among their patients. Therefore, continuous monitoring of vaccine perception among the HCWs is pertinent to a successful vaccination program and safeguards the healthcare system. There is a rapid and tremendous growth of literature regarding COVID-19 vaccine hesitancy and acceptance in HCWs globally, leading to the need to update the community and government on these issues.

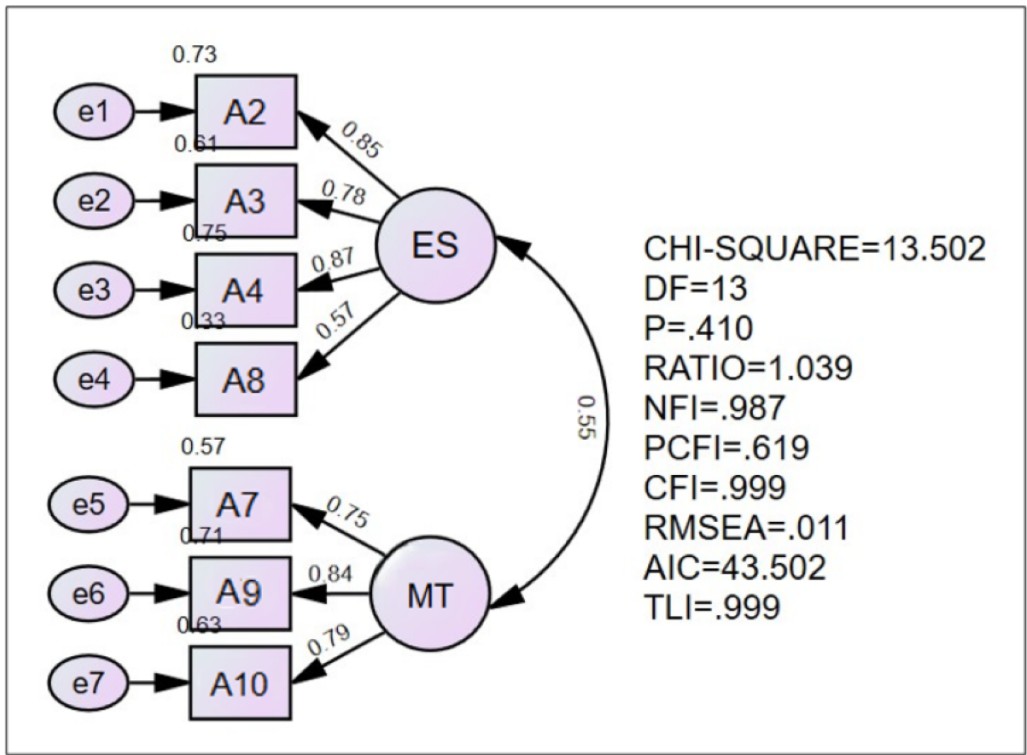

**Figure 1  Final CoVaP model for healthcare workers.** Note. *Domain ES refers to efficacy and safety, and Domain MT refers to misinformation and trust issues.

**Table 3  Results of factor loadings, AVE and CR for the CoVaP.**

| Items | Standardised factor loadings | Domains | [a]Cronbach's alpha | [b]AVE | [b]CR |
|---|---|---|---|---|---|
| A2 | 0.85 | | | | |
| A3 | 0.78 | Efficacy & Safety (ES) | 0.899 | 0.603 | 0.856 |
| A4 | 0.87 | | | | |
| A8 | 0.57 | | | | |
| A7 | 0.75 | Misinformation & Trust Issues (MT) | 0.815 | 0.631 | 0.836 |
| A9 | 0.84 | | | | |
| A10 | 0.79 | | | | |

Notes.
[a] Reliability analysis; Cronbach's Alpha Coefficient, overall Cronbach's alpha = 0.873.
[b] AVE and CR were calculated manually based on the formula given by *Fornell & Larcker (1981)*.

Many studies have been conducted on the perceptions of COVID-19 vaccination using various instruments, but it has not been validated properly (*Fares et al., 2021*; *Mejia et al., 2021*; *Sirait et al., 2024*). Hence, this study aimed to develop and perform a detailed validation of an instrument that can evaluate the perceptions of the COVID-19 vaccine among HCWs in Malaysia. Fortunately, this will help bridge the gap, as the findings might be valuable for other studies.

The CoVaP instrument demonstrated high content and face validity, with item-level content validity index (I-CVI) scores ranging from 0.83 to 1.0 and a scale-level content validity index (S-CVI/Ave) of 0.97. Similarly, the face validity indices (I-FVI and S-FVI/Ave) were also high, indicating that the items were relevant, clear, and comprehensible to the target population. These findings suggest that the CoVaP instrument is well-suited for assessing HCWs' perceptions of COVID-19 vaccination. EFA and CFA supported a two-factor model comprising seven items, categorized into two domains: efficacy and safety (ES) and misinformation and trust issues (MT).

Interestingly, the items that were developed, especially based on the occupational risk of HCWs, were removed during the analysis, probably indicating that the perceptions of HCWs regarding the COVID-19 vaccine were similar to those of the general public (*Wong et al., 2023*). This indicates that most HCWs in Malaysia did not perceive that the occupational risk of getting COVID-19 infection is an important issue, nor do they believe that they must take the vaccine due to occupational risk. These results must be interpreted carefully, and further studies are needed to explore this issue. Undermining the issue of occupational risk among HCWs is very important to safeguard preventive measures in future pandemics. There could be several variables contributing to this, such as their lack of involvement in the clinical setting or their pre-existing mindset of being more susceptible to contracting infections in the community than clinical work. Therefore, it is advisable for healthcare facility management to consistently disseminate information about the high susceptibility of HCWs to COVID-19 infection due to the nature of their job.

Domain efficacy and safety (ES) consists of items related to confidence in the vaccine and risk calculation aspect. This domain consists of items on "I believe the COVID-19 vaccine is safe because it has been clinically tested.", "I feel the effectiveness of the vaccine outweighs the risk of serious side effects" and "I think this vaccine protects us from the severe complications of COVID-19 infection". These results demonstrated that the 5C model theoretical framework can be used to explain the perceptions of HCWs in this study, and this model has been used extensively in studies on the perception of COVID-19 vaccination (*Rancher et al., 2023*). The 5C model postulates five level determinants of vaccine hesitancy: confidence, complacency, constraints, risk calculation, and collective responsibility (*Nuwarda et al., 2022*). Confidence refers to trust in the vaccine's safety and efficacy; complacency relates to perceptions of the infection as a danger and whether vaccination is required; constraints include structural and psychological barriers related to vaccination intention and uptake; risk calculation refers to comparing the risk of infection *versus* immunization. Finally, collective responsibility involves the desire and willingness to become vaccinated to protect others or to generate herd immunity (*Rancher et al., 2023*).

While trust issues about vaccine safety and efficacy were significant predictors of hesitancy globally (*McCready et al., 2023*), CoVaP results suggest that Malaysian HCWs showed higher vaccine confidence and low risk of infection (risk calculation). The item related to distrust in the severe side effect reports issued by the Malaysian Ministry of Health was removed during the analysis. This is probably because the HCWs in Malaysia have better access to real-world efficacy data and trust in the authorities. This highlights the importance of local trust in government policies, setting the Malaysian context apart

from findings in Western studies, where individual autonomy and lack of trust in the vaccine, vaccine producers, or the government often shaped the barriers to the COVID-19 vaccination (*Goodwin et al., 2022*; *Rancher et al., 2023*).

However, the items for religious issues, vaccine-containing electronic chips and using complementary medicines to protect against COVID-19 infection were similar to other studies conducted among HCWs (*McCready et al., 2023*). Malaysia is a multi-religious country with a diverse population that practices various faiths, and Muslims constitute the majority of the population. Thus, religious issue also plays an important factor in shaping the perception of vaccines among HCWs. This was similar to studies conducted in countries that are predominantly Muslim (*Abu Farha et al., 2021*; *Fares et al., 2021*) and also other studies in Malaysia (*Lau et al., 2021*; *Syed Alwi et al. 2021*). Understanding these factors is essential for designing effective public health interventions to address vaccine hesitancy and improve vaccination rates.

The CoVaP instrument has demonstrated robust psychometric properties for assessing HCWs' perceptions of COVID-19 vaccination. Given its strong content and construct validity, as well as its reliability, there is potential for this instrument to be adapted for use with other vaccines and in different epidemic contexts. The CoVaP instrument's core domains—efficacy and safety (ES) and misinformation and trust issues (MT)—encompass fundamental aspects of vaccine acceptance and hesitancy that are not specific to COVID-19. For instance, concerns about the efficacy and safety of vaccines are universal and apply to any vaccination campaign. The items in the CoVaP instrument that address these concerns can be easily adapted to assess perceptions of other vaccines, such as those for influenza, measles, or HPV. Misinformation and trust issues are common barriers to vaccine acceptance across different diseases. The CoVaP instrument's items related to misinformation and trust can be modified to reflect the specific context of other vaccines, making it a versatile tool for assessing vaccine perceptions. During various epidemics, the rapid development and deployment of vaccines often lead to similar challenges in public perception. By changing the items to reflect the particular disease and vaccine in issue, the CoVaP instrument can be modified to meet these situations.

There are several limitations in our investigation. The study respondents were chosen *via* chain-referral sampling, which introduces the possibility of selection bias. However, this method is to ensure that the anonymity and privacy of the respondents are carefully safeguarded. This is done because the topic is considered a sensitive issue and may influence the working environment of the respondents. Hence, the research was conducted not using official channels. Test-retest reliability was not conducted in this study due to time constraints. We used different time frames and contact resources for EFA and CFA data collection to safeguard that those HCWs in the EFA and CFA are different. Besides, we also restricted the Google Form to be filled out once only. However, we could not control the outcomes since they may be repetitive to the findings.

Moreover, our study encompassed HCWs employed across diverse units within public and private healthcare facilities, an aspect that has yet to be extensively investigated in Malaysia. This study's research subject is contemporary and significant, although it has yet to get much attention in Malaysia. Despite these constraints, the study answers the

question. Therefore, the findings may be utilized as a source of information for developing interventional strategies that aim to reduce vaccine hesitancy and increase vaccination rates among healthcare workers, who are the gatekeepers in the healthcare system. Consequently, this can significantly impact the efforts to prevent and control COVID-19 inside the nation.

## CONCLUSION

The CoVaP instrument is a reliable and valid tool for assessing HCWs' perceptions of COVID-19 vaccination in Malaysia. With excellent validity, as well as strong reliability, the CoVaP instrument effectively captures the unique cultural and contextual factors influencing vaccine perceptions among Malaysian HCWs. Its adaptability for use with other vaccines and in different epidemic contexts further enhances its value, making it a valuable resource for public health authorities and researchers to design targeted interventions, reduce vaccine hesitancy, and improve vaccination uptake among HCWs, ultimately strengthening the healthcare system's response to public health challenges.

## ACKNOWLEDGEMENTS

We would like to extend our heartfelt gratitude to all the participants who contributed their time and insights to this study. We also wish to acknowledge the assistance of AI tools, specifically Grammarly and Copilot, for their support in refining the manuscript.

### Funding
The work was supported by the Ministry of Higher Education Malaysia's Fundamental Research Grant Scheme (Ref: FRGS/1/2021/SKK04/USM/01/1). The funders had no role in study design, data collection and analysis, decision to publish, or preparation of the manuscript.

### Grant Disclosures
The following grant information was disclosed by the authors:
Ministry of Higher Education Malaysia's Fundamental Research Grant Scheme: FRGS/1/2021/SKK04/USM/01/1.

### Competing Interests
Norhayati Mohd Noor is an Academic Editor for PeerJ.

### Author Contributions
- Siti Nur Aisyah Zaid conceived and designed the experiments, performed the experiments, analyzed the data, prepared figures and/or tables, authored or reviewed drafts of the article, and approved the final draft.
- Azidah Abdul Kadir conceived and designed the experiments, performed the experiments, analyzed the data, prepared figures and/or tables, authored or reviewed drafts of the article, and approved the final draft.

- Mohd Noor Norhayati conceived and designed the experiments, prepared figures and/or tables, authored or reviewed drafts of the article, and approved the final draft.
- Basaruddin Ahmad conceived and designed the experiments, analyzed the data, prepared figures and/or tables, authored or reviewed drafts of the article, and approved the final draft.
- Muhamad Saiful Bahri Yusoff conceived and designed the experiments, authored or reviewed drafts of the article, and approved the final draft.
- Anis Safura Ramli conceived and designed the experiments, authored or reviewed drafts of the article, and approved the final draft.
- Jasy Suet Yan Liew conceived and designed the experiments, authored or reviewed drafts of the article, and approved the final draft.

## Human Ethics

The following information was supplied relating to ethical approvals (i.e., approving body and any reference numbers):

The Universiti Sains Malaysia Research and Ethical Committee granted Ethical approval for the study (Ethical Application Ref: USM/JEPeM/21100700).

## Field Study Permissions

The following information was supplied relating to field study approvals (i.e., approving body and any reference numbers):

The field study was approved by the Malaysia Ministry of Health (project number: NMRR ID-21-02113-ZGG (IIR)).

## Data Availability

The data set of this study is available in the Supplemental Files and at: https://opendata.usm.my/handle/123456789/74707.

## Supplemental Information

Supplemental information for this article can be found online at http://dx.doi.org/10.7717/peerj.19318#supplemental-information.

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
