# Peer review of "Development and validation of COVID-19 vaccination perception (CoVaP) instrument among healthcare workers in Malaysia"

_PeerJ, doi:10.7717/peerj.19318_

## Round 0.1 · original submission · Major Revisions

I have read the feedback from each of the reviewers and recommend that you revise your manuscript in line with their comments and the PeerJ editorial criteria: https://peerj.com/about/editorial-criteria/.

Reviewer 1 ·

Basic reporting

This research seeks to design and verify the COVID-19 Vaccination Perceptions (CoVaP) questionnaire among Malaysian health-care workers.The article is written in clear, straightforward, and technically accurate English. adhere to professional norms of civility and articulation.The article's structure adheres to an accepted format of 'standard parts.'
Figure pertinent to the article's topic, with enough resolution, and suitably described and labeled.

Experimental design

Methods articulated with enough detail to enable reproducibility by another researcher for further validation and enhancement of the questionnaire of COVID-19 Vaccination Perception (CoVaP) Instrument Among Healthcare Workers

Validity of the findings

The measure has seven elements, categorized into two domains: effectiveness and safety; disinformation and trust. Items were derived from a literature review of current tools and insights from public health and vaccination specialists, so ensuring the content validity of the instrument from the outset of the project. The findings demonstrated a significant degree of content validity and face validity for CoVaP.

Reviewer 2 ·

Basic reporting

The manuscript is written in clear and professional English, with no major grammatical errors. However, minor editing could further enhance readability.The study references recent and relevant literature, the structure adheres to PeerJ’s guidelines with standard sections. Figures are relevant. Raw data files are included.

Experimental design

The manuscript clearly defines the research question.The study uses established validation techniques.
The methodology is described with sufficient detail for replication.

Validity of the findings

The manuscript fulfills the outlined criteria.Data are comprehensive and statistically sound. The conclusions are clearly connected to the research question and are supported by strong, well-executed analysis.

Additional comments

No additional comments

Reviewer 3 ·

Basic reporting

The reporting was mostly clear with professional language. However, the literature cited did not represent the depth of knowledge in the reseach topic which indicated sub-optimal literature review process

Experimental design

Major flaws were found including reliance on papers to extract the novel survey items; these papers were not valid

Validity of the findings

Data were collected two years ago. Vaccine hesitancy is known to be time-specific phenomenon. Thus, the results of the study would be outdated.

Additional comments

Thanks for the invitation to review this manuscript.
In the current study, Siti Nur Aisyah Zaid, Azidah Abdul Kadir et al. developed and tested the validity of a novel survey instrument to assess the attitude toward COVID-19 vaccination among health professionals in Malaysia.
Significant flaws were detected in the methods, literature review, and clarity that would undermine the contribution of this word to the research topic.
I have the following specific points:
1. In the Abstract, please indicate the language(s) of the survey instrument developed. Malay only? Or both Malay and English with translation and back-translation?
2. In the Introduction, lines 66 – 68: Please rephrase the sentence “The revised definition of vaccine hesitancy now …”. The revision is needed since this is the standard definition of vaccine resistance/hesitancy.
3. In the Introduction, lines 68 – 73: This introductory section on the determinants of vaccination hesitancy appears very short and it is unclear whether it refers to COVID-19 or vaccination in general. I assume that the authors were referring to the determinants of COVID-19 vaccine hesitancy based on the reference cited “Vaccine Sentiments in Malaysia: Narratives of Comments from Facebook Post”. I recommend starting with a brief introduction on the determinants of vaccine hesitancy in general with reference to well described theoretical frameworks specifically designed for this purpose (e.g., the 5C model, the 7C model, the vaccine hesitancy scale (VHS) model, and the vaccine conspiracy beliefs scale (VCBS) model). Then, the authors can briefly provide specific results on how these models were used during the COVID-19 pandemic to assess the attitude to COVID-19 vaccination among various population demographics including health professionals.
4. In the Introduction, lines 73 – 75: The authors’ justification of selection of HCWs as the study groups does not appear sufficient “The impact of vaccine hesitancy is expected to be greater and pertinent to public health particularly when involving vulnerable frontline groups such as healthcare workers (HCWs).” I recommend adding more information regarding the role of health professionals to improve health literacy, recommend vaccination, be a source of infection to fragile or vulnerable groups, professional losses from absenteeism, etc. Also, please provide specific examples from the literature to support your selection of HCWs as the study group.
5. In the Introduction, the authors can briefly mention an overview regarding the status of COVID-19 vaccination in Malaysia, the study setting. (E.g., the reported rates of COVID-19 vaccine acceptance among the general population and among HCWs, mandatory vaccination policies against COVID-19, is the vaccine currently provided free of charge, what types of COVID-19 vaccines were used in the country, etc.)
6. In the Abstract, lines 36 – 37: The sentence “Content … who rated the item, average, and instrument-face validity.” is not totally clear. Please consider revising these this sentence. A suggestion, “who assessed content and face validity”.
7. In the Introduction, line 87: the authors stated that “limited valid instruments are available”; however, in the Abstract, lines 27 – 28: they stated that “valid instruments for assessing the HCWs’ perceptions of COVID-19 vaccination were lacking”. Please resolve this discrepancy.
8. In the Introduction, lines 87 – 88: Please explain the exact intended meaning of the word “multicultural”. How is it relevant to vaccine hesitancy in Malaysia?
9. In the Methods, lines 107 – 111: this justification of the need of a specific instrument to assess COVID-19 vaccine hesitancy for Malaysian participants only lacks any credible evidence. Also, “a review of the literature was conducted” is not sufficient. Who conducted the review, in what scientific databases, what did you search for (exact keywords, MeSH terms, time the search was concluded)
10. Lines 107 – 111: This was the major caveat of the study since I can point out to several different published papers coming from Malaysia on COVID-19 vaccine hesitancy that were not considered by the authors which points to suboptimal process of literature review (a few examples below):
A. Lau JFW, Woon YL, Leong CT, Teh HS. Factors influencing acceptance of the COVID-19 vaccine in Malaysia: a web-based survey. Osong Public Health Res Perspect. 2021 Dec;12(6):361-373. doi: 10.24171/j.phrp.2021.0085. Epub 2021 Nov 25. PMID: 34818501; PMCID: PMC8721269.
B. Kalok A, Razak Dali W, Sharip S, Abdullah B, Kamarudin M, Dasrilsyah RA, Abdul Rahman R, Kamisan Atan I. Maternal COVID-19 vaccine acceptance among Malaysian pregnant women: A multicenter cross-sectional study. Front Public Health. 2023 Feb 22;11:1092724. doi: 10.3389/fpubh.2023.1092724. PMID: 36908400; PMCID: PMC9992805.
C. Syed Alwi SAR, Rafidah E, Zurraini A, Juslina O, Brohi IB, Lukas S. A survey on COVID-19 vaccine acceptance and concern among Malaysians. BMC Public Health. 2021 Jun 12;21(1):1129. doi: 10.1186/s12889-021-11071-6. PMID: 34118897; PMCID: PMC8196915.
D. Mohd Anuar AH, Mohamad Anuar NN, Isa SNI, Bahari M. The level of knowledge and acceptance towards the COVID-19 vaccine among the community in Johor Bahru, Johor. Med J Malaysia. 2024 Mar;79(Suppl 1):88-95. PMID: 38555891.
E. Hing NYL, Woon YL, Lee YK, Kim HJ, Lothfi NM, Wong E, Perialathan K, Ahmad Sanusi NH, Isa A, Leong CT, Costa-Font J. When do persuasive messages on vaccine safety steer COVID-19 vaccine acceptance and recommendations? Behavioural insights from a randomised controlled experiment in Malaysia. BMJ Glob Health. 2022 Jul;7(7):e009250. doi: 10.1136/bmjgh-2022-009250. PMID: 35906015; PMCID: PMC9344599.
11. Another major methodological flaw is the reliance on studies that did not use well-tested or validated survey instruments as the basis for your selection of the items for the novel instrument. Additionally, among these five references, at least 3 were used for the general public rather than HCWs.
12. Another limitation of the study is that the data were collected two years ago. Vaccine hesitancy is known to be time-specific phenomenon. Thus, the results of the study would be outdated.
13. Line 177: the use of 0.3 as the factor loading cut-off for retaining items needs justification since the common standard used is 0.4
14. The Discussion and Conclusion sections were very brief and did not elaborate on the study results based on the extensive literature on COVID-19 vaccine hesitancy. Several key relevant references were missing indicating that the authors do not have the depth of knowledge needed to work in this research field.

Reviewer 4 ·

Basic reporting

- (page 4, abstract) Please give the meaning of COVID-19, I-CVI, I-FVI, and S-FVI acronyms on the first mention. Also don’t forget to mention the statistics used to analysed the data.
- (page 6, intro) This sentence: “The coronavirus disease 2019 (COVID-19) has caused a substantive impact on social, psychological, economic, health, and mortality globally” will be more significant if it’s also supported by another relevant study. For example, https://doi.org/10.52225/narrax.v1i1.71
- Authors are suggested to proofread the manuscript after addressing all comments to avoid any typological, grammatical, and lingual mistakes and errors. For example, “A few item items” line 120; “the 14th. August” line 147.
- (page 11) This sentence “Many studies have been conducted on the perceptions of COVID-19 vaccination using the instrument, but it has not been validated properly.” also shares the same idea with a similar study by Sirait et al. https://pubmed.ncbi.nlm.nih.gov/38798859/ Kindly include this reference to make the sentence stronger.

Experimental design

- (line 200) what is the p-value used in this study?

Validity of the findings

- (page 12) More literature is required regarding implications of study findings.
- I suggest that the authors rewrite the conclusions part, as it does not align well with the findings obtained in the study and is somehow too short.

·

Basic reporting

no comment

Experimental design

Phase 2: Development of the Scale

132 Content Validity: why N=7?
Can this number (N=7) of reviewers be considered sufficient and representative of the rest of the opinions in the medical sector?

Construct Validity and Reliability
169 The data collection for EFA took place from August 20th to September 20th, 2022,
170 while the data collection for CFA was carried out from October 20th to November 20th, 2022. (What is the data collection period? Only the starting period has been specified.)

Validity of the findings

no comment

Additional comments

The researchers aimed to demonstrate the role of healthcare workers in facilitating COVID-19 vaccination by assessing healthcare workers’ perceptions of COVID-19 vaccination to increase their confidence in the vaccination processes. The aim was to develop and validate a COVID-19 vaccination perception tool among healthcare workers in Malaysia.
In principle, I can see the effort that researchers have put into this work, but I feel that limiting it to COVID-19 vaccines may be too outdated. Therefore, it may be useful to note the possibility of developing this method to fit with vaccines in general, especially in cases of different epidemics, without limiting it to a specific disease.

---

## Round 0.2 · accepted · Accept

After the revisions, I am pleased to accept your manuscript for publication.

Reviewer 4 ·

Basic reporting

no comment

Experimental design

no comment

Validity of the findings

no comment

Additional comments

no comment

·

Basic reporting

no comment

Experimental design

no comment

Validity of the findings

no comment

Additional comments

I am satisfied with this manuscript after the recent modifications.